# Influencing Factors and Realization Path of Power Decarbonization—Based on Panel Data Analysis of 30 Provinces in China from 2011 to 2019

**DOI:** 10.3390/ijerph192315930

**Published:** 2022-11-29

**Authors:** Ning Ren, Xiufan Zhang, Decheng Fan

**Affiliations:** 1College of Marxism, Zhejiang Sci-Tech University, Hangzhou 310018, China; 2School of Economics and Management, Zhejiang Sci-Tech University, Hangzhou 310018, China; 3School of Economics and Management, Harbin Engineering University, Harbin 150006, China

**Keywords:** power decarbonization, power generation, supply chain, power system, technology innovation

## Abstract

2011–2019 was the critical period of the low-carbon transformation of the power industry, reflecting the deepening influence of market mechanisms. Decarbonization of the new power system is a systematic project that needs to strengthen the top-level design and overall planning. Therefore, the paper first evaluates the decarbonization of the existing power system and controls the grid architecture, power structure, energy utilization, supply chain, and trading market to further optimize the system by strengthening the basic theoretical research of the new power system, exploring the key elements of the low-carbon development of the power system, promoting the breakthrough of the key subjects, and formulating the new power system decarbonization path. In the international push for carbon neutrality goals, identifying key factors in the decarbonization of the power system is critical to achieving low-carbon development in the power sector. Combined with the characteristics and development trends of the power industry, the five dimensions of “Power generation decarbonization (SP)”, “Energy utilization efficiency (EU)”, “Supply chain decarbonization (SC)”, and “Power grid decarbonization (PG)”, and “the Trading system (TS)” are selected to construct an evaluation index system for the power decarbonization and identify the key factors. The Analytic Network Process (ANP) Method is used to calculate the index weight and measure the decarbonization level of the power industry in 30 provinces in China from 2011 to 2019. The evaluation results reveal that the overall decarbonization level of the power industry is on the rise and has stabilized after peaking in 2016. The regression results of the systematic GMM estimation show that “the intensity of cross-regional transmission”, “the degree of carbon market participation”, “technology innovation”, and “policy support” can significantly promote power decarbonization, and different regions have heterogeneity. Therefore, we propose to achieve technological innovation and upgrading in the eastern region, strengthen the construction of smart grids in the central region, optimize the power structure in the western region, and improve the market mechanism as a whole, to form a low-carbon development path for the power industry.

## 1. Introduction

Since 2021, the worldwide energy shortage issue has become increasingly severe. According to the “2022 electricity market report” released by the International Energy Agency (IEA), global electricity demand experienced a slight decline in 2020 and ushered in growth in 2021. In 2021, global electricity demand rose by six percent, half of which came from China. China’s electricity demand is tremendous, the proportion of coal-fired power generation is high, and the cleanness degree of the power industry is relatively low [1,2,3]. With the continuous increase in the total power generation, the proportion of non-fossil energy has reached 34.5% in 2021, which makes the total carbon emissions of the power industry remain high.

With the proposal of international carbon neutrality goals, 127 countries have fulfilled their commitments. In the future, China’s power system will closely combine energy conservation, efficiency improvement, and carbon emission reduction targets [4]. The clean development of the power industry has become an important engine for the carbon neutrality goal. The power industry is one of the key industries of carbon emission control [5]. However, the supply and demand of power resources are inversely distributed, resulting in higher transportation costs and higher energy losses [6]. The power system in the EU (European Union) is faced with an unreasonable power grid structure during the use of new energy [7]. In the process of large-scale grid connection, the stable operation and energy balance of the power grid is still facing great challenges. The power systems in the EU still require coping with the challenge of a high proportion of renewable energy consumption. At the same time, the new coronavirus disease and various epidemic prevention measures have forced economic activity to reduce or be suspended, which has a certain impact on the power supply, the power grid operation, the power market transactions, and other aspects. The new power system is an important carrier for promoting the low-carbon transformation of the power industry.

Relative research has focused on the introduction of technical means in the power supply and power grid, which proves that technological upgrading can promote carbon emission reduction [8]. Power Decarbonization requires access to a high proportion of new energy. It is difficult for the traditional power system to encounter the requirements of an energy-efficient operation. The power system is complex and has the characteristics of a network structure [9]. It is composed of multiple participants and investors, forming an input-output system with energy production and consumption as the main functions. The construction of a power system needs to fully realize the interaction and coordinated evolution of the power supply, the power grid, the supply chain, and other parts [10,11]. Through systematic analysis, the key factors of low-carbon development of the power industry have formed rich results. However, from the overall perspective, the construction of a new power system and the overall low-carbon development path of the power industry need to be further explored. The innovation of this paper is to clarify the key influencing factors in the process of low-carbon development of the power system, verify the support provided by technology, market, and policy for the low-carbon development of the power system through the model, and reveal the realization path of the power industry theoretically and empirically based on the development basis of different regions. China actively improves the efficiency of the power system. In the power system, the interaction and influence of multiple participants and investors is a complex input-output system with network structure characteristics. Therefore, the study objectives are to manage the low-carbon of the power system from the perspective of the system and to identify the key problems of low-carbon operation efficiency of the power system. The purpose of this paper is to promote the construction of a clean and low-carbon power system. The main contribution is to identify the problems, including the utilization rate of thermal power equipment and the low load rate of the power grid. To establish the system GMM model and put forward the corresponding solutions. Explore the establishment of a power transmission mechanism that supports multi-domain coordination of power grids, power sources, energy conservation, supply chains, and emerging market players to improve new low-carbon power systems.

Based on the above analysis, the structure of this paper is arranged as follows: The second part is the literature review. The third part is the research method. From the five dimensions of ‘Power generation decarbonization’, ‘Energy utilization efficiency’, ‘Supply chain decarbonization’, ‘Power grid decarbonization’, and ‘The Trading system’, we construct the evaluation index system of low-carbon development of the power system. The fourth part is the evaluation results. The fifth part is the discussion. We establish the GMM model of the low-carbon development of the power industry in 30 provinces of China and clarify the realized path through empirical analysis. The sixth part is the research conclusion and prospects.

## 2. Literature Review

For the long-term planning scenario, power system decarbonization is necessary. By summarizing the research results of low-carbon development of the power industry, some scholars have pointed out that power supply planning, the introduction of trading mechanisms, the use of new energy technologies, and the construction of smart grids can achieve the low-carbon development of the power industry. However, during the operation of the power system, there are still decarbonization bottlenecks. We review the key elements and decarbonization measures as follows.

### 2.1. The Bottleneck of Low-Carbon Operation of the Power System

#### 2.1.1. High Carbon Emission of Power Generation

Fragkos et al. (2017) [12] believed that power generation decarbonization is the key to improving the energy efficiency of coal-fired power generation. However, under the growing demand for power generation, the proportion of traditional energy used in conventional power sources is relatively high. The clean capacity of thermal power is rather low [13,14]. Coal-fired generating units are installed with environmental protection facilities; however, the mediation process of energy-saving power generation dispatching will generate stop losses, bring variable costs, generate higher operation and maintenance costs, and increase the cost of power generation enterprises. Fang et al. (2022) [15] used the data from 42 thermal power plants in China in 2020 and found that input redundancy, high carbon emission intensity of power supply, and heating are the main reasons for low carbon emission efficiency. The resource allocation and input-output structure of the power supply should be adjusted to achieve carbon emission reduction in the power industry. The change in power generation structure is an important method to reduce carbon emission intensity. Sun et al. (2021) [16] used the data envelopment analysis (DEA) method to measure wind power efficiency in 30 provinces in China from 2012 to 2017 and pointed out that improving wind power efficiency will help China achieve energy conservation and emission reduction targets. While vigorously advancing new energy, the development of conventional energy should be coordinated. The technological upgrading of thermal power should be accelerated.

#### 2.1.2. Energy Efficiency Loss

Higher conversion rates of power generation and consumption can reduce losses in the power transfer process, thereby reducing total carbon emissions. In the process of power conversion, it faces the loss of energy efficiency. Due to the relative lack of primary energy, self-sufficient power generation in some regions of China is difficult to meet the electricity consumption [17]. Strengthening the construction of transmission channels can soar the intensity of cross-regional clean energy transmission, improve transmission efficiency, and promote the cleanness of energy supply with new energy. Mier et al. (2020) [18] used the model of the European electricity market to study and found that under the carbon emission reduction target, energy efficiency contributed 11% of carbon emission reductions, and intermittent renewable energy, such as wind energy and solar energy, accounted for 53%. Therefore, improving energy efficiency is the key issue of electricity market decarbonization. Cross-regional power transmission ensures the timely integration and consumption of new energy, which is conducive to improving energy utilization efficiency in distribution links and production.

#### 2.1.3. Weak Integration Capability of the Supply Chain

The integration of the supply chain aims at realizing low-carbon planning in the power industry [19]. At present, the production and operation efficiency of the power supply chain in China needs to be improved. Huang et al. (2021) [20] established a non-radial two-stage model to evaluate the sustainable performance of China’s provincial power supply chain system (PSCS), pointing out that effective management and technical systems are conducive to carbon emission reduction. The inspection ability of the power grid material is weak, and the ability to swallow large-scale storage clusters has not been formed, which is not conducive to the intelligent deployment of power sources and the monitoring of equipment transportation. Based on the Stackelberg model framework, Wang et al. (2021) [21] introduced the total carbon quota control and trading conditions of supply chain enterprises, constructed a carbon emission reduction coordination game model for coal-fired power supply chain enterprises, and designed a revenue-sharing coordination mechanism among supply chain enterprises. It is found that the coordinated development of the supply chain is an important step and link in the development of the carbon trading market, and it is also a necessary guarantee to implement the overall goal of carbon emission reduction in China. Therefore, the supply chain should provide strong support for the demand for power grid construction and the construction of energy-saving demonstration areas.

#### 2.1.4. Low Efficiency of Power Transmission

The power grid is the central link and core hub of the low-carbon transformation for the power industry. It links power production and consumption to achieve high efficiency, reduction, and electrification [22]. At present, there is a loss in the process of regional dispatching and transmission, which increases the demand for power generation and the transmission efficiency of the power grid needs to be improved. China plans to prioritize the dispatch of clean energy, increase the proportion of clean energy, and realize energy substitution and intelligence [23]. The smart grid is upgraded to displace the high energy consumption and reduce the energy consumption in the transmission process [24]. Eberle and Heath (2020) [25] pointed out that the modernization of the power grid is conducive to the effective allocation of resources to reduce carbon emissions.

#### 2.1.5. No Linkage between the Electricity Price and the Carbon Price

The power industry is the focus of the carbon emissions trading mechanism. With the continuous expansion of the carbon trading market into power enterprises, the allocation of resources in the power industry should be further optimized through the market mechanism [26,27,28]. At present, in the regional and national parallel carbon emission trading market, the total carbon emission target of the power industry is mainly set with the lowest joint cost as a single target, ignoring the emission reduction effect of specific links in power production [29,30]. In the context of the macroeconomic downturn and the outbreak of new corona pneumonia, the unit-load decreases, and the coal consumption for power generation increases, which is not conducive to the sustainable development of the power industry.

### 2.2. Key Breakthrough Elements of Low Carbon Transformation Development

#### 2.2.1. Technological Innovation

Innovation of technology is the key means for the power industry to achieve low carbon goals. According to the type of technology, technological innovation can be further subdivided:Energy technology innovation.

Energy technology is important to realize the reuse of carbon resources from multiple aspects. First, from the perspective of power generation, energy technology innovation can improve the grid-related performance of generating units. By developing large-capacity, high-density, high-security, and low-cost energy storage devices, energy technology innovation can promote efficient utilization of clean energy. Second, carbon capture and storage technologies can improve the energy efficiency of electricity. Third, the energy technology innovation in power grid planning, construction, operation, and maintenance is conducive to coordinating the conventional power supply and large-scale power grid operation. Sovacool and Monyei (2021) [31] studied the importance of technology configuration in the development of the power industry in China, the European Union, India, and the United States. By setting scenarios, it is pointed out that energy technology innovation can reduce carbon emissions, and potential low-carbon options can provide higher positive externalities.

2.Blockchain technology innovation.

Blockchain technology is mainly applied to the power grid. By relying on blockchain technology, the power grid can integrate energy internet resources and coordinate clean development and system costs [32]. The blockchain is used to establish a distributed grid; then, the technical adaptability, business scalability, and system operation level of the grid are improved. The region covers the data confirmation and transaction links of the power industry and realizes the distributed data storage and tamper-proof modification, thus forming data asset protection [33,34].

3.Digital technology innovation.

Digital technology encourages the intelligent collaborative management of the power industry and realizes the bidirectional improvement of production efficiency. Artificial intelligence technology enables the traditional power grid to improve the accuracy of the production process [35]. The new energy cloud platform can improve the early warning level of the power grid to deal with natural disasters, external damage, and equipment defects by applying big data, effectively guaranteeing the safe operation of energy use. Digital technology helps to improve the production process, thereby continually enhancing the ability of resource allocation and flexible interoperability and ameliorating the efficiency of power operation and maintenance. Pietzcker et al. (2021) [30] found that the use of fossil carbon capture and storage (CCS) can significantly reduce cumulative carbon emissions in the power sector, easing the electricity burden on the industrial sector without increasing electricity prices or total system costs.

#### 2.2.2. Carbon Trading Market Maturity

The global carbon emission trading market develops fast and covers a large number of industries, aiming at reducing the overall cost of carbon emission reduction. The allocation mechanism sets the upper limit for the enterprises and sends a carbon emission reduction signal to the enterprise. Wu et al. (2020) [36] used the two-step goal programming allocation model to allocate the quota of carbon emission for the power industry. They analyze the impact of carbon emission quota allocation on the electricity market, propose a two-stage algorithm, and establish a multi-objective optimization model that minimizes the total cost and carbon emission cost of the power system. Pietzcker et al. (2021) [30] point out that the European Union advocates carbon neutrality by 2050 and thus has stricter market discipline targets. By fitting the marginal abatement cost curve of the power sector, it is found that the carbon emission reduction target will accelerate the transformation of the power system by 3–17 years. The operation of the carbon trading market and electricity trading market shows a trend of improvement. The organic combination of electricity price and carbon emission cost is conducive to the coordination of the carbon trading market and power market, improving the market competitiveness of clean energy, and promoting the low-carbon transformation of energy.

#### 2.2.3. Policy Support

The government gives policy support and financial support to environmental-friendly power enterprises, thus helping them to further improve R&D capabilities [37]. Zappa et al. (2021) [38] conducted a scenario simulation of the evolution of power systems in Central and Western Europe under the Paris Agreement and designed electricity markets to achieve low-carbon development goals under policy constraints. On the one hand, the government’s financial support power enterprises to increase technological R&D efforts. Khanna et al. (2021) [39] examined the impact of the US National Renewable Portfolio Standard (RPS) climate policy on greenhouse gases and welfare, assessing carbon reductions and welfare costs. The study found that when policy standards are in the right range, they can improve carbon reduction without weakening welfare; on the other hand, in the economic fluctuation environment, the policy support is conducive to enterprises against the economic cycle and reduce its negative impact. Financial and policy support can provide relatively stable external environment support for the sustainable development of power enterprises.

## 3. Materials and Methods

In this section, we set the index system to realize the low-carbon development of the power system. We aim at building a new power system with new energy as the main body, forming a low-carbon development as the basic premise, the smart grid as the hub platform, the open interaction between the power supply and the grid as the driving force, and the power market and the carbon trading market as the support. Since 2011, China has established a pilot market for carbon emission trading in seven provinces or cities and incorporated the power industry into key industries, setting carbon emission quotas to achieve low-carbon operation goals. In 2015, China ushered in the second reform of the power system to achieve electricity marketization. Two rounds of power system reform have promoted the transformation of traditional power generation enterprises and the upgrading of the power industry. In 2019, China proposed to continue to step up the pace of power system reform, improve efficiency, reduce costs, optimize resource allocation, and start a national unified carbon emissions trading market. The year 2011–2019 is the key period of low-carbon transformation of the power industry, reflecting the deepening influence of market mechanisms. Therefore, we study the low-carbon process of the power industry from 2011 to 2019 to further explore the realized path.

A flowchart of the experiment is shown in Figure 1.

### 3.1. Evaluation Index Selection

We summarize the construction method of the evaluation index system. Some scholars choose to subjectively screen the evaluation indicators based on expert experience, while some scholars use quantitative methods such as principal component analysis to objectively screen the indicators. The problem of expert subjective screening is strong subjective randomness, while the problem of objective screening is over-reliance on indicator data, ignoring the actual meaning of indicators. Therefore, firstly, according to the bottlenecks and key elements encountered in the process of decarbonization in the power industry, the index selection is carried out. On this basis, the variation coefficient is used to quantitatively screen the indicators from Power generation decarbonization (SP), Energy utilization efficiency (EU), and Supply chain decarbonization (SC). Power grid decarbonization (PG) and The Trading system (TS) establish 5 criterion layers and select indexes in a large range. According to the principle of maximum information content and elimination of redundant information, the evaluation index system of decarbonization of variation and the indexes are quantitatively screened to construct the evaluation index system of decarbonization in the power industry.

#### 3.1.1. Selected Evaluation Index in a Large Range

Summarize the bottlenecks and key elements encountered in the development of the power industry and the large-scale screening of evaluation indicators based on the high-frequency influencing factors listed by domestic and foreign scholars on the low-carbon development of the power industry. According to the literature review, the bottlenecks of low-carbon operation of the power industry include high carbon emission of power generation, efficiency loss, weak integration ability, low transmission efficiency, and no linkage between the electricity price and the carbon price. Therefore, in the construction of a new power system, these factors are considered. The evaluation index system includes 5 primary indicators, which represent ‘Power generation decarbonization (SP)’, ‘Energy utilization efficiency (EU)’, ‘Supply chain decarbonization (SC)’, ‘Power grid decarbonization (PG)’, and ‘The Trading system (TS)’. Each primary indicator contains several secondary and tertiary indicators. According to the literature review, the index system of the five criterion layers, including 75 indexes such as ‘‘The scale of thermal power (SP11)’’ was selected.

#### 3.1.2. Index Screening Principle of Maximum Information Content

The information content of the index is solved by the coefficient of variation, and the index with the largest information content in the same category is selected to ensure that the selected index has a significant impact on the decarbonization of the power industry. The coefficient of variation of the index reflects the identification ability of the index in the decarbonization evaluation of the power industry. The greater the coefficient of variation is, the greater the distribution variability of the index in the evaluation of decarbonization in the power industry is; the greater the information content is, the stronger the information resolution of the index is; on the contrary, the weaker, it should be eliminated, so as to screen out the indicators that have the greatest impact on the decarbonization evaluation of the power industry. Finally, five criterion layers are constructed by Power generation decarbonization (SP), Energy utilization efficiency (EU), Supply chain decarbonization (SC), Power grid decarbonization (PG), and The Trading system (TS). The green industry evaluation index system, including 27 indicators such as water consumption reduction of CNY 10,000 of the GDP, makes the selected indicators have the largest information content and has a significant impact on the decarbonization evaluation of the power industry.

Set v_i_ as the coefficient of variation of index *i*, *n* as the number of evaluated objects, ${\bar{x}}_{i}$ as the mean value of data in each year of index *i*, and *x_it_* as the value of index *i* in the *t* year.
$$v_{i}=\frac{\sqrt{\frac{1}{n}\sum_{i=1}^{n}{\left(x_{it}-{\bar{x}}_{i}\right)}^{2}}}{{\bar{x}}_{i}}$$
$${\bar{x}}_{i}=\frac{1}{n}\sum_{i=1}^{n}x_{it}$$

Solve the coefficient of variation of the indicators of Secondary indicators and Tertiary indicators, and retain the top two or three indicators of the coefficient of variation in each level, to screen out the indicators with the largest information content.

(1)Power generation decarbonization (SP).

The index reveals the low carbonization of power generation. It contains 3 secondary indicators: ‘Thermal power generation (SP1)’, ‘Clean energy generation (SP2)’, and ‘Photovoltaic power generation (SP3)’. ‘Thermal power generation (SP1)’ includes two tertiary indicators: ‘The scale of thermal power (SP11)’ and ‘Optimization of industrial structure (SP12)’, [40,41]. ‘Clean energy generation (SP2)’ includes 2 tertiary indicators: ‘The scale of wind power (SP21)’ and ‘The scale of integration (SP22)’ [42,43]. ‘Photovoltaic power generation (SP3)’ includes two tertiary indicators: ‘Solar photovoltaic industry scale (SP31)’ and ‘Photoelectric conversion efficiency (SP32)’ [44].

(2)Energy utilization efficiency (EU).

The index reflects the utilization and transformation level of energy in the power industry. It includes 2 secondary indicators: ‘Energy-saving generation scheduling (EU1)’ and ‘Power conversion (EU2)’ [45]. ‘Energy-saving generation scheduling (EU1)’ includes three tertiary indicators: ‘Unit start-stop loss (EU11)’, ‘Replacement electricity income (EU12)’, and ‘Fixed cost compensation (EU13)’ [46,47]. ‘Power conversion (EU2)’ includes 3 tertiary indicators: ‘The conversion rate of thermal power fuel (EU21)’, ‘Power generation conversion effect (EU22)’ and ‘Power consumption intensity (EU23)’, [48,49].

(3)Supply chain decarbonization (SC).

The index reflects the low-carbon level of power operation, product, and the value chain, including 2 secondary indicators: ‘Carbon source flow (SC1)’ and ‘Integrated operation capability (SC2)’. ‘Carbon source flow (SC1)’ includes 2 tertiary indicators: Byproduct recovery rate (SC11) and ‘Carbon emission recovery rate (SC12)’, [50]. ‘Integrated operation capability (SC2)’ includes 4 tertiary indicators: ‘Power grid material inspection (SC21)’, ‘Storage cluster throughput (SC22)’, ‘Green procurement (SC23)’ and ‘Quality control ability (SC24)’.

(4)Power grid decarbonization (PG).

The index reflects the low-carbon level of power transportation, including 2 secondary indicators: ‘Energy consumption at the supply and demand side (PG1)’ and ‘Power grid construction (PG2)’, [51,52]. ‘Energy consumption at the supply and demand side (PG1)’ includes 2 tertiary indicators: ‘The coal consumption rate of power generation (PG11)’ and ‘The coal consumption rate of power generation (PG12)’, [53,54], which reflects whether the store transportation can achieve high efficiency, reduction, and electrification. ‘Power grid construction (PG2)’ includes 2 tertiary indicators: ‘Smart grid construction (PG21)’ and ‘Grid Loss (*PG22*)’, [55,56], reflecting power transmission efficiency.

(5)The Trading system (TS).

The index reflects the support of the market trading mechanism for the power industry, including 2 secondary indicators: Carbon quota allocation (TS1) and Market effectiveness (TS2), [29,57]. Carbon quota allocation (TS1) includes the compact scale of quota allocation (TS11) and Carbon emission data statistics and verification (TS12). Market effectiveness (TS2) includes 3 tertiary indicators: The linkage between Electricity Price and Carbon Market Price (TS21), The volatility of electricity price (TS22), and Electricity market size (TS23). The tighter quota allocation setting and accurate carbon emission data statistics are conducive to the emission reduction incentive effect of the trading mechanism. At the same time, price linkage and large market scale are conducive to deepening the reform of the power market to help low carbon transformation.

#### 3.1.3. Rationality Judgment of Index System

Referring to the research of Shi and Chi (2014) [58], a reasonable index system uses less than 30% of the indicators to reflect more than 95% of the original information, and the index system is considered reasonable. The variance of the original data of the final index system is compared with the variance of the original data of the selected index system, which is the information content of the index system.

Set *S* as the covariance matrix of the indicator data; *trS* is the trace of the covariance matrix; *s* is the number of indicators after screening; *h* is the number of election indicators. The information contribution rate in of the selected index to the sea selection index is:*In = trS_s_/trS_h_*

That is, the sum of variances *trS_s_* of *s* indicators after screening. The ratio of *trS_s_* to the sum of the variances of the *h* selected indicators represents the information of the *h* selected indicators reflected by the s selected indicators.
*In* = *trS_s_/trS_h_* = 2.5236 × 10^15^/2.6685 × 10^15^ = 94.57%

By screening 36% (27/75 = 36%) indicators, 94.57% of the original information of the sea selection index system is reflected, which meets the requirements of the reasonable index system.

### 3.2. Measurement of Evaluation Index

In this part, we further measure the three-level indicators. Among them, the data on thermal power, clean energy power generation, and photovoltaic power generation are derived from the ‘regional energy balance Table’ in the ‘China Energy Yearbook’, and the data of coal consumption rate in the energy consumption of the power supply are derived from the ‘Local Statistical Yearbook’. All kinds of energy are converted into 10,000 tons of standard coal equivalent. The data on energy-saving generation dispatching, power conversion, and power grid construction are from the ‘China Electricity Yearbook’. The data on carbon quota allocation and market effectiveness are from the ‘China Carbon Emission Quota Allocation Scheme’, ‘China Carbon Trading Market Depth Survey Report from 2021 to 2025’, and ‘China Electricity Yearbook’. With 2000 as the base period, the regional GDP is converted into a constant price.

Considering that it is difficult to use accurate data for quantitative analysis of the indicators under the index of ‘carbon source flow’ and ‘integrated operation capability in the ‘low-carbon supply chain’. These indexes are divided into five levels by using the Likert scale, which is ‘very low, low, moderate, high, and very high’, and the corresponding scores are ‘0, 0.25, 0.5, 0.75, 1’. The experts are invited to score. Thus, the evaluation index system of the low-carbon level of the power industry can be formed (Table 1).

### 3.3. Index Weight Calculation

Due to the five dimensions of ‘‘Power generation decarbonization (SP)’’, ‘‘Energy utilization efficiency (EU)’’, ‘‘Supply chain decarbonization (SC)”, ‘‘Power grid decarbonization (PG) ‘‘ and ‘‘the Trading system (TS)” have mutual linkage effects between layers, and the indicators of each layer have certain common characteristics. If the evaluation methods such as the analytic hierarchy process, fuzzy comprehensive evaluation method, or grey correlation analysis method are used, this problem may not be eliminated. At the same time, these methods have certain subjectivity in the process of assigning weights to indicators, which may affect the reliability of the evaluation index system. The analytic network process (ANP) is a decision-making method based on the analytic hierarchy process (AHP), which has the characteristics of multi-attribute evaluation and feedback. At the same time, Super Decision software (Thomas L. Saaty and William Adams, Pittsburgh, PA, America) can be used to process the supermatrix of the multi-element set, which is simple to calculate. Therefore, we use the Analytic Network Process (ANP) Method and Super Decision software to evaluate the effect of decarbonization in the power industry

#### 3.3.1. Intrinsic Analysis of Indicators

In the network structure, the primary layer is the low-carbon development level of the power industry, and the indicators in the second layer and the tertiary layer are interactive and interconnected.

First, we study the feedback or dependence between the indicators. We designed a two-dimensional form questionnaire to collect the correlation of indicators identified by experts. The principles for the selection of expert teams are as follows: experts shall be included in the list of experts in the power industry expert pool published by the provincial energy bureau, who shall be specialized in power engineering and shall serve as professors and academicians or senior engineers. According to the list of selected experts, some experts are from the Institute of Electrical Engineering of the Chinese Academy of Sciences, the Chinese Academy of Electric Power Sciences, and the State Grid Institute of Electric Power Sciences. They have a full understanding of the power industry reform, resource conservation, and development prospects. Some experts are directly involved in the low-carbon transformation and construction of China’s power industry. In March 2021, we conducted a questionnaire survey among 53 experts, including 15 experts in thermal power enterprises, 12 experts in hydropower enterprises, 11 experts in new energy enterprises, 2 experts in nuclear power enterprises, 2 experts in power transmission and transformation enterprises, and 11 experts in power supply. Using the Delphi method, Experts are invited to give comprehensive consideration to the operation of China’s power industry, the development of the carbon trading market and the trading situation of the power market, and the correlation between the indicators according to their own experience and understanding. In the first questionnaire, experts disagreed about the interrelationship between SP3 and EU2. Five experts believed that ‘Photovoltaic power generation (SP3)’ and ‘Power conversion (EU2)’ should be independent. The opinions of all experts were summarized. Through three repeated operations, the experts agreed that there was a correlation between the two. The interrelationship between the elements was pairwise compared to form an indicator Table, as shown in Table 2:

#### 3.3.2. Construction of Judgment Matrix

Experts are invited to score pairwise comparison evaluation indexes according to the ‘1-9 scale method’ to construct the judgment matrix, as shown in Table 3.

By sending an email, the issue of comparison between the evaluation indexes of low-carbon development level of the power industry is sent to each expert separately. All expert opinions are collected and anonymously fed back to each expert and then consulted again. Through six centralized feedbacks, unanimous opinions are finally obtained and pass the consistency test.

#### 3.3.3. Construction of Super Decision Matrix

Given the large scale of the matrix formed by the ANP method, the matrix calculation part is omitted. The weight of the super limit matrix is calculated by ‘Super Decision software’. According to the two-level and three-level comparison matrices provided above, the node cluster is created. The correlation between the node cluster and the nodes is determined by the ‘correlation degree table’. The judgment matrix is input, and the supermatrix, the weighting matrix, and the limit supermatrix are calculated. The elements are sorted according to priority, and the factors that play a major role are identified. Finally, we obtain the weight of each index.

## 4. Results

### 4.1. Weight Calculation Results

Due to the different attributes and dimensions of each observation index, the average method is used to remove dimensions. The min–max normalization method is a common method to normalize the original data. The original data matrix is set as *O* = (*o_ij_*)*_mn_*, and the normalized matrix is *x* = (*x_ij_*)*_m*n_*, (*i* = 1, 2, …, *m*; *j* = 1, 2, …, *n*). The conversion formulas are shown as follows:

For positive indicators:(1)$$x_{ij}=\frac{o_{ij}-\min(o_{j})}{\max(o_{j})-\min(o_{j})},i=1,2,3,\ldots ,n$$

For negative indicators:(2)$$x_{ij}=\frac{\min(o_{j})-o_{ij}}{\max(o_{j})-\min(o_{j})},i=1,2,3,\ldots ,n$$

The weight of the ultra-limit matrix is calculated by ‘Super Decision software’, and the weight of the tertiary layer indexes is obtained by weighting. In summary, the weights of the tertiary indexes are obtained, as shown in Table 4.

It can be seen from Table 4 that the weight of indicators such as ‘The coal consumption rate of power generation’, ‘Power consumption intensity’, ‘Power generation conversion effect’, ‘Smart grid construction’, and ‘The clean Energy Structure of Power Grid’ have a great impact on the low-carbon development level of the power industry. The weight level of the indexes for low carbon development of the power industry can be obtained, as shown in Figure 2.

The results of the evaluation of the index weights can be seen to be consistent with the method of Haghi et al. (2020) [19] for finding the optimal decarbonization of electricity and heating systems in the United Kingdom. Power transportation and power grid construction determine the cost-effectiveness and low carbon of the electrical system. A comprehensive approach should be adopted for long-term planning to ensure cost-effective decarbonization.

### 4.2. Evaluation Results

We select the panel data on the low-carbon development of the power industry in 30 provinces of China from 2011 to 2019 and calculate their low-carbon development score of them from 2011 to 2019 according to the standardized index data and index weight. According to the distribution of China’s administrative regions, 30 provinces are divided into eastern, central, and western regions. The average scores of cities in different regions are calculated to evaluate the development of the power industry in different regions and different dimensions, as shown in Figure 3, Figure 4 and Figure 5.

In the eastern region, the decrease in the proportion of thermal power generation (i.e., the change of power generation structure) is an important factor in promoting the improvement of the low-carbon development score of power generation. The power grid score in the eastern region has increased rapidly in recent years, but the score is still less than 0.5, with larger room for improvement (Figure 3). Taking East China Power Grid in the Yangtze River Delta as an example, the light load of 500 KV transmission lines is relatively serious, and most of them are under long-term light load.

In the central region, the low-carbon scores showed a slight upward trend in fluctuations, the overall energy efficiency of China’s power industry fluctuates, and the energy efficiency in the western region is slightly higher than that in other regions (Figure 4). The reason is that the line loss rate of China’s power industry is slightly higher than that of advanced international power companies. China’s power equipment technical parameters are relatively backward; supercritical units accounted for only 4.1% of the total installed capacity of thermal power, while the United States, Japan, and Russia have accounted for more than 50%. The equipment levels of new energy and renewable energy need to be further improved. In recent years, due to the slow growth of electric power, the degree of redundancy is still rising. The integrated line loss rate of China’s power grid ranks at the advanced level of the countries with the same power supply load density in the world, but there is still a gap compared with the countries with the lowest loss rate. This result coincides with the conclusion that Bonilla et al. (2022) [59] proposed the use of renewable energy to achieve full decarbonization of the electricity market after studying the Spanish power industry. At the same time, it can further extend its research results and improve the use of new energy in the process of power grid construction.

In the western region, the current low-carbon score of China’s power industry supply chain is high (Figure 5). In recent years, China has focused on peak shaving and the valley filling of electricity demand, dealing with the situation of power shortage, and continuously improving the utilization rate of power, but it still needs to further improve the efficiency of terminal power consumption.

The score of the market system development shows a rapid increase in China (Figure 3, Figure 4 and Figure 5). At present, the regional electricity market in southern China has started trial operations, covering Guangdong, Guangxi, Yunnan, Guizhou, Hainan, and the other five provinces. The electricity price is adjusted according to the actual supply and demand situation to avoid the large fluctuation in electricity prices. This marks the accelerated construction of a unified national electricity market system. The role of the market in the optimal allocation of resources has been significantly enhanced, and the varieties and compensation mechanism of the auxiliary service market will be further improved.

Thus, it is a complex task to determine the optimal combination of key elements for the country’s new power system, thus applying it to the long-term decarbonization process in China’s electricity market.

## 5. Discussion

After evaluating the low-carbon development status of the power industry in 30 provinces in China, we further explore the path of the new power system and discuss the situations in different regions. We construct the system GMM to identify the path of low-carbon development in the power industry. The instrumental variables are used to solve the endogenous problem between the explained variables and some explanatory variables. Through the heterogeneity analysis, we make suggestions for the basis and potential of power development in different regions.

### 5.1. System GMM Model

Taking the evaluation results of the index system of the low-carbon development of the power industry as the dependent variable. By reviewing and summarizing the classical literature, the key breakthrough factors in the process of low-carbon development of the power system are extracted, which are used as independent variables to construct the following model:*LDI_it_* = *α*_0_ + *α*_1_*TIAR_it_* + *α*_2_*CM_it_* + *α*_3_*RI_it_* + *α*_4_*BTI_it_* + *α*_5_*DT_t_* + *α*_6_*PS_t_* + *α**CONS* + *ε*(3)
where

*i*: the province.

*t*: the year.

*LDI*: the low-carbon development level of the power industry.

*TIAR*: the intensity of cross-regional transmission, which is measured by the proportion of regional power generation and power consumption.

*CM*: the degree of carbon market participation, which is measured by the ratio of the number of electric power enterprises incorporated into the pilot market to the total number of electric power enterprises.

*RI*: R&D investment, which is measured by the ratio of R&D investment to regional GDP.

*BTI*: blockchain technology innovation, which is measured by the provincial big data development index.

*DT*: digital technology, which is measured by the supporting investment in new infrastructure construction.

*PS*: policy support, which is measured by loans to support the power industry.

Control variables: Environmental regulation intensity (*SER*), measured by the ratio of environmental governance costs and the total output value of the power industry; The level of regional economic development (*RE*), measured based on GDP per capita in 2000; Foreign direct investment *(FDI*), measured by the proportion of regional FDI in GDP.

The system GMM estimation regression results are shown in Table 5.

There is a significant correlation between the explanatory variables and the low-carbon level of the power industry (Table 5). The coefficients of the *TIAR* and *BTI* variables are significantly positive (at the 1% level). The improvement of the power trans-regional transmission level and the innovation of blockchain technology can promote the low-carbon development of the power industry. This conclusion further extends the research results of Mar et al. (2021) [60] on power optimization models for long-term planning scenarios. Therefore, it is essential to further develop blockchain technology and tap the development potential of blockchain technology. The coefficients of the variables of *CM*, *RI*, *DT*, and *PS* are significantly positive at the 5% level. Market participation, R&D, and policy support have a positive impact on the low-carbon transformation in the development of the power industry. The power generation technology proposed by Broesicke et al. (2021) [14] and Haghi et al. (2020) [19] can better solve the conflict between cogeneration and decarbonization targets. While using clean energy, it is necessary to further improve energy utilization and conversion efficiency and achieve clean production by enhancing the synergy of the supply chain. Gyanwali et al. (2021) [22] considered hydrogen and CCS technologies to achieve deep decarbonization of the integrated power grid and further referred to their research conclusions for smart grid construction. Bodal et al. (2020) [13] also pointed out the joint planning of electricity and hydrogen production to achieve decarbonization synergies. The government’s low-carbon policy fully plays the role of guiding the low-carbon transformation of the power industry.

### 5.2. Endogenous Analysis

The low-carbon development level of the power industry may have a reverse causal relationship with energy technology innovation and cross-regional transportation. At the same time, the measurement error of variables and the omission of important variables will cause endogenous errors in the estimation results. We use instrumental variables to solve this problem.

Firstly, the number of power researchers (SP) is used as a tool for R&D investment.

Secondly, the number of companies with blockchain business (DRCE) is used as a tool for blockchain technology innovation variable.

Finally, the number of power transmission lines (RPL) is used as the instrumental variable of cross-regional transmission intensity.

The results of IV-GMM estimation are shown in Table 6.

After considering endogeneity, each explanatory variable still has a positive impact on the low-carbon development level of the power industry. The instrumental variables are reasonable. Each variable exerts a positive part in the low-carbon development level of the power industry, and the influence of each variable is consistent.

### 5.3. Robustness Test

First, we replace the explained variable. The decline in the total carbon emissions of the power industry is used as the explained variable. Second, we replace the measurement method of explanatory variables. Replace the policy support variables with tax and investment variables. Third, we include the one-period lag and two-period lag of each explanatory variable in the regression model. The results in Table 7 show that the robustness analysis results consist of the baseline model estimation results.

### 5.4. Heterogeneity Analysis

It can be seen from the evaluation index system that scores in different regions of China vary significantly in different regions. Therefore, we conduct regional heterogeneity analysis to identify the low-carbon development path of the power industry in different regions. The samples are divided into eastern, central, and western regions, and the system GMM estimation regression results are shown in Table 8.

The results of Table 8 show that the variable coefficients of digital technology and policy support in the eastern region are higher, the variable coefficients of cross-regional transport, R&D investment, and blockchain technology innovation in the central region are higher, and the variable coefficients of cross-regional transport and blockchain technology innovation in the western region are higher. Carbon trading market variables can play a significant impact in the three regions. Therefore, the low-carbon development paths of the power industry in different regions are summarized as follows:

#### 5.4.1. Technological Innovation and Upgrading in the Eastern Region

The eastern region should further strengthen the use of digital technology, increase regional infrastructure investment, and optimize the power generation structure. Speed up the digital upgrade of power systems, comprehensively promote the application of new technologies, and bring innovation to the operation mode of the power industry. From Figure 3, we know that the eastern region should further strengthen the construction of the power grid. Through technological innovation and upgrading, the construction of a smart grid can greatly improve the efficiency of decarbonization (Figure 3). Government support is an important path, which can be formed from various aspects of tax, investment, and financial support policies. In conclusion, the market performs a decisive function in resource allocation, and the government forms a supporting role.

#### 5.4.2. Smart Grid Upgrades in the Central Region

The central region should soar the intensity of cross-regional transmission, accelerate the construction of the UHV backbone network in central China, erect a platform for optimal allocation of clean resources, and achieve the acceptance of clean energy. Accelerate the improvement of blockchain technology innovation ability, and improve its ability to absorb new energy. Figure 4 shows that the energy efficiency of the power sector in the central region is relatively low, which can be improved by establishing a smart grid to help the power grid upgrade to energy internet., accelerating the construction of information collection, perception, processing, and promoting the sharing of various data.

#### 5.4.3. Optimization and Upgrading of the Power Structure in the Western Region

The low-carbon development path of the power industry in the western region and the central region is relatively similar. It is necessary to improve the power cross-regional transmission and realize the technological innovation of blockchain. It can be seen from Figure 5 that the low-carbon development of the power industry in the western region is relatively good, and the upgrading and optimization of the power structure can be further considered. For power transmission in the western region, it is necessary to improve the main network in the northwest region and accelerate the construction of the network. Ensure that clean energy is connected to the grid, and create a clean energy optimal allocation platform. Promote the application of power substitution technology. Speed up the construction of auxiliary power supply, promote the establishment of long-term transmission mechanism, and optimize the power supply.

#### 5.4.4. Improving Market Mechanisms

The development of the carbon trading market in various regions can play a significant impact on the low-carbon power industry. China should further establish a coordination mechanism between the power market and the carbon market, establish a relationship between the carbon price and the electricity price, and form a pricing system. By adopting a flexible pricing rule, the construction of big data such as power generation, consumption, and cross-regional transmission is strengthened to encourage the participation of clean energy in spot trading. Under the background of the rapid construction of the national carbon trading market, the market should further play the role of expanding consumption space and realize the aim of “low-carbon “by market-oriented means.

## 6. Conclusions

China’s decarbonization of new power systems and the growing demand for supply make the need for comprehensive planning of power systems increasingly prominent. In this paper, the evaluation index system of power decarbonization is constructed to identify the key factors of power decarbonization. From the weight results of the evaluation index system, the decarbonization of the power system is largely due to the grasp of the Grid load rate, Grid Line Loss Rate, Power generation conversion effect (EU22), and Power consumption intensity (EU23) throughout the planning period, and further attention should be paid to the intermittence of wind and solar energy and the change of load. The study objective is to construct the path to realize the low-carbon development of the power industry, and the following conclusions are obtained: (1) The key factors in the process of power decarbonization include the coal consumption rate of power generation, Power consumption intensity, ‘Power generation conversion effect’, ‘Smart grid construction’, and ‘The clean Energy Structure of the Power Grid’. PG12, EU23, EU22, PG21, EU13 (2) The overall development level of low-carbon in China’s power industry is on the rise, and it maintains a relatively stable trend. There are regional differences in different dimensions of the power system. Due to the regional difference, the eastern region should further improve energy efficiency, the central region should improve energy efficiency and construct a smart power grid, and the western region should strengthen the construction of the power supply chain. The eastern region should concentrate on technological innovation and upgrading, the central region should focus on the upgrading of the smart grid, and the western region should focus on the optimization and upgrading of the power structure. From the overall perspective, the study result shows that we should quicken the construction of the national carbon emissions trading market and power market, form dual support from the market and government, and build an important path to realize the low-carbon development of China’s power industry. (3) The system GMM estimation regression results show that the key path of new power system construction is to improve the technical standards and capabilities, from the overall enhance the transmission and consumption of clean energy power capacity. Through the optimization and adjustment of power supply configuration and market operation, the operation of the transmission network and power system is optimized, the coordination mechanism of transmission and consumption combined with the intergovernmental agreement and power market is improved, and the market mechanism adapted to the new power system is improved. Through the above path, the decarbonization development of the new power system is explored.

In this paper, the research scenario is defined as the construction of a new type of power system in China with the goal of low-carbon development. In the process of constructing the index system, some indicators are not selected. We emphasize the research of low-carbon development status, influencing factors, emission reduction paths, and processes of multi-agent. There is a lack of characterization of the supply chain and power generation structure of the power industry in detail and a more detailed analysis of the influencing mechanism of the path to achieving low carbon in different regions. Therefore, in the future, we can further study how to better highlight the utility of new energy in different dimensions, reduce cost and increase efficiency, and measure the decarbonization contribution of the power industry.

## Figures and Tables

**Figure 1 ijerph-19-15930-f001:**
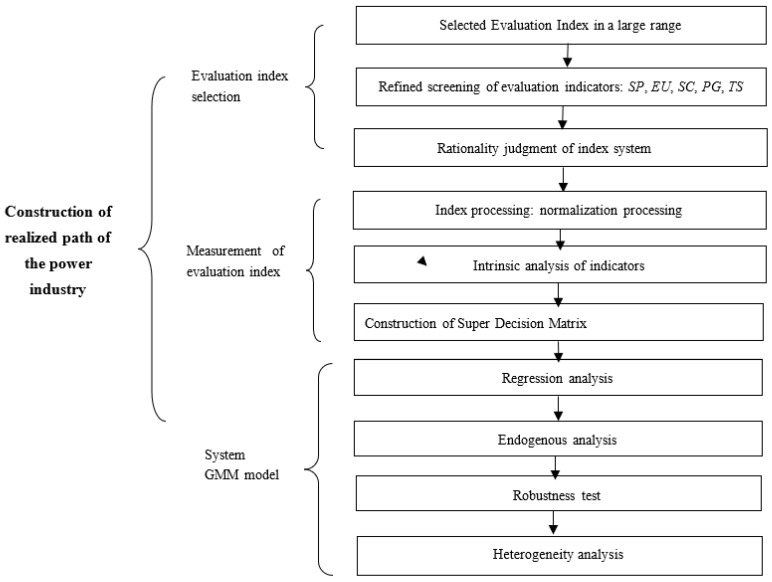
A Flowchart of the Experiment.

**Figure 2 ijerph-19-15930-f002:**
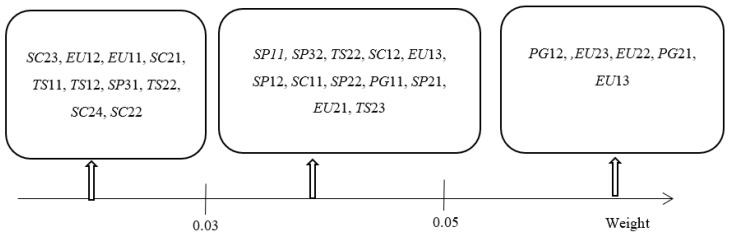
Weight Level Diagram of Evaluation Indexes for Low Carbon Development of the Power Industry.

**Figure 3 ijerph-19-15930-f003:**
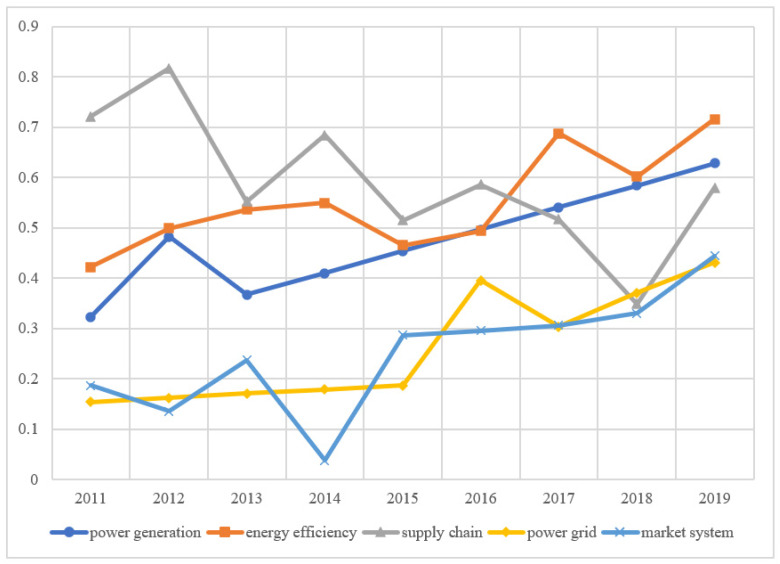
Evaluation score of low-carbon development in eastern China.

**Figure 4 ijerph-19-15930-f004:**
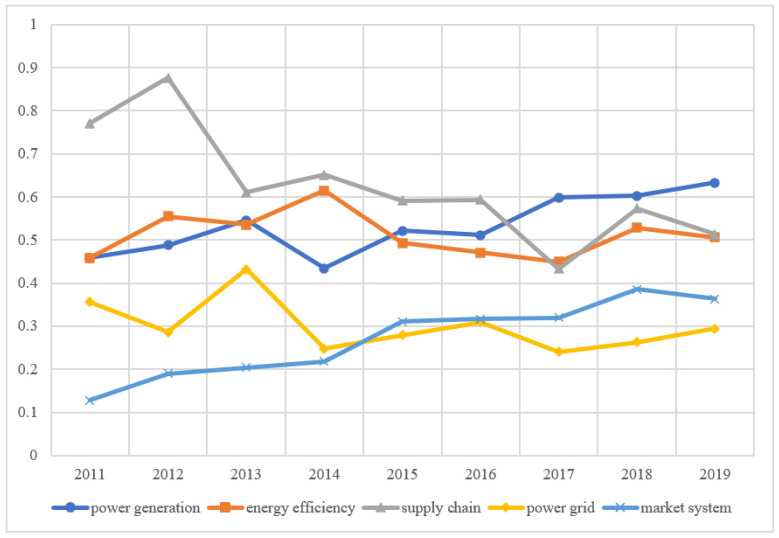
Evaluation score of low-carbon development in central China.

**Figure 5 ijerph-19-15930-f005:**
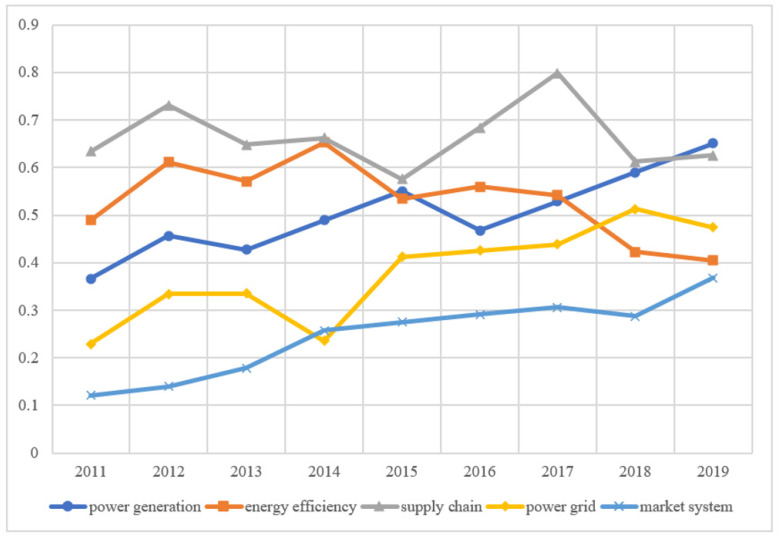
Evaluation score of low-carbon development in western China.

**Table 1 ijerph-19-15930-t001:** Evaluation index system of the low-carbon level of the power industry.

low-carbon level of the power industry	**Primary Indicators**	**Secondary Indicators**	**Tertiary Indicators**	**Indicator Interpretation**	**Attributes**
	*SP*	Thermal power generation (*SP*1)	The scale of thermal power (*SP*11)	The utilization hours of thermal power equipment	Reverse
			Optimization of industrial structure (*SP*12)	The scale of High Efficiency and Large Capacity Units	Positive
		Clean energy generation (*SP*2)	The scale of wind power (*SP*21)	The proportion of installed capacity	Positive
			The scale of integration (*SP*22)	Fan connection rate	Positive
		Photovoltaic power generation (*SP*3)	Solar photovoltaic industry scale (*SP*31)	The proportion of photovoltaic installed capacity	Positive
			Photoelectric conversion efficiency (*SP*32)	The conversion efficiency of solar photovoltaic cells	Positive
	*EU*	Energy-saving generation scheduling (*SP*1)	Unit start-stop loss (*EU*11)	Phased loss estimates for start-stop processes	Reverse
			Replacement electricity income (*EU*12)	Marginal income of generating units	Positive
			Fixed cost compensation (*EU*13)	Fixed cost compensation under energy-saving generation dispatching	Reverse
		Power conversion (*EU*2)	The conversion rate of thermal power fuel (*EU*21)	The conversion efficiency of coal fuel consumption for power generation	Positive
			Power generation conversion effect (*EU*22)	The proportion of regional power generation and power consumption	Positive
			Power consumption intensity (*EU*23)	Electricity consumption per unit GDP	Positive
	*SC*	Carbon source flow (*SC*1)	Byproduct recovery rate (*SC*11)	The recovery efficiency of by-products in the power generation process	Positive
			Carbon emission recovery rate (*SC*12)	The recovery rate of carbon emission	Positive
		Integrated operation capability (*SC*2)	Power grid material inspection (*SC*21)	Power grid material quality detection capability	Positive
			Storage cluster throughput (*SC*22)	Storage cluster throughput capacity of the power supply chain	Positive
			Green procurement (*SC*23)	Green purchasing ability of the power supply chain	Positive
			Quality control ability (*SC*24)	Power Supply Chain Control Capability	Positive
	*PG*	Energy consumption at the supply and demand side (*PG*1)	The coal consumption rate of power generation (*PG*11)	Standard coal consumption per cycle/power generation per cycle	Reverse
			The coal consumption rate of power generation (*PG*12)	Standard coal consumption per cycle/power generation per cycle	Reverse
		Power grid construction (*PG*2)	Smart grid construction (*PG*21)	Grid load rate	Positive
			Grid Loss (*PG*22)	Grid Line Loss Rate	Reverse
	*TS*	Carbon quota allocation (*TS*1)	The compact scale of quota allocation (*TS*11)	The ratio of Total Quota to Total Carbon Emissions of Regional power Enterprises	Positive
			Carbon emission data statistics and verification (*TS*12)	The construction of regional carbon emission data information disclosure platform	Positive
		Market effectiveness (*TS*2)	The linkage between Electricity Price and Carbon Market Price (*TS*21)	The correlation coefficient between regional electricity price and carbon price	Positive
			The volatility of electricity price (*TS*22)	The volatility of regional electricity price	Positive
			Electricity market size (*TS*23)	Actual power consumption data of the regional power market	Positive

**Table 2 ijerph-19-15930-t002:** We form an indicator table according to the interrelationship between the elements, which were pairwise compared. The index interrelationship of the evaluation index system criterion layer is as follows.

Normal Level Index	*SP*	*EU*	*SC*	*PG*	*TS*
SP	18	15	10	8	12
EU	16	14	13	7	13
SC	12	10	8	10	10
PG	10	8	6	5	8
TS	12	10	8	10	12

**Table 3 ijerph-19-15930-t003:** We use the Delphi method to construct the judgment matrix.

*f_ij_*	Meaning
1	*f_i_* is as important as *f_j_*
3	*f_i_* and *f_j_* are slightly important
5	*f_i_* and *f_j_* are important
7	*f_i_* and *f_j_* are quite important
9	*f_i_* and *f_j_* are extremely important
2, 4, 6, 8	Between 1–3, 3–5, 5–7, and 7–9, respectively

**Table 4 ijerph-19-15930-t004:** Evaluation index system of the low-carbon level of the power industry.

Primary Indicators	Secondary Indicators	Tertiary Indicators	Indicator Interpretation	Weights
*SP*	Thermal power generation (*SP*1)	The scale of thermal power (*SP*11)	The utilization hours of thermal power equipment	0.033
		Optimization of industrial structure (*SP*12)	The scale of High Efficiency and Large Capacity Units	0.037
	Clean energy generation (*SP*2)	The scale of wind power (*SP*21)	The proportion of installed capacity	0.044
		The scale of integration (*SP*22)	Fan connection rate	0.042
	Photovoltaic power generation (*SP*3)	Solar photovoltaic industry scale (*SP*31)	The proportion of photovoltaic installed capacity	0.025
		Photoelectric conversion efficiency (*SP*32)	The conversion efficiency of solar photovoltaic cells	0.032
*EU*	Energy-saving generation scheduling (*SP*1)	Unit start-stop loss (*EU*11)	Phased loss estimates for start-stop processes	0.021
		Replacement electricity income (*EU*12)	Marginal income of generating units	0.020
		Fixed cost compensation (*EU*13)	Fixed cost compensation under energy-saving generation dispatching	0.035
	Power conversion (*EU*2)	The conversion rate of thermal power fuel (*EU*21)	The conversion efficiency of coal fuel consumption for power generation	0.046
		Power generation conversion effect (*EU*22)	The proportion of regional power generation and power consumption	0.061
		Power consumption intensity (*EU*23)	Electricity consumption per unit GDP	0.058
*SC*	Carbon source flow (*SC*1)	Byproduct recovery rate (*SC*11)	The recovery efficiency of by-products in the power generation process	0.038
		Carbon emission recovery rate (*SC*12)	The recovery rate of carbon emission	0.034
	Integrated operation capability (*SC*2)	Power grid material inspection (*SC*21)	Power grid material quality detection capability	0.030
		Storage cluster throughput (*SC*22)	Storage cluster throughput capacity of the power supply chain	0.022
		Green procurement (*SC*23)	Green purchasing ability of the power supply chain	0.013
		Quality control ability (*SC*24)	Power Supply Chain Control Capability	0.028
*PG*	Energy consumption at the supply and demand side (*PG*1)	The coal consumption rate of power generation (*PG*11)	Standard coal consumption per cycle/power generation per cycle	0.051
		The coal consumption rate of power generation (*PG*12)	Standard coal consumption per cycle/power generation per cycle	0.042
	Power grid construction (*PG*2)	Smart grid construction (*PG*21)	Grid load rate	0.066
		Grid Loss (*PG*22)	Grid Line Loss Rate	0.069
*TS*	Carbon quota allocation (*TS*1)	The compact scale of quota allocation (*TS*11)	The ratio of Total Quota to Total Carbon Emissions of Regional power Enterprises	0.023
		Carbon emission data statistics and verification (*TS*12)	The construction of regional carbon emission data information disclosure platform	0.024
	Market effectiveness (*TS*2)	The linkage between Electricity Price and Carbon Market Price (*TS*21)	The correlation coefficient between regional electricity price and the carbon price	0.027
		The volatility of electricity price (*TS*22)	The volatility of regional electricity price	0.032
		Electricity market size (*TS*23)	Actual power consumption data of the regional power market	0.047

**Table 5 ijerph-19-15930-t005:** Regression analysis results.

Variable	GMM	OLS	FE
*TIAR*	0.2144 *** (0.0871)	0.1493 *** (0.0243)	0.1449 *** (0.0349)
*CM*	0.1621 *** (0.0000)	0.1342 *** (0.0217)	0.0994 ** (0.0274)
*RI*	0.1850 ** (0.0410)	0.1572 ** (0.0464)	0.1405 *** (0.0147)
*BTI*	0.1361 ** (0.0289)	0.1250 ** (0.0334)	0.1051 ** (0.0471)
*DT*	0.2465 ** (0.1699)	0.0915 * (0.0622)	0.1674 *** (0.0097)
*PS*	0.1766 ** (0.1824)	0.1147 * (0.0814)	0.1039 * (0.0821)
*CONS*	0.1689 ** (0.0318)	0.1013 *** (0.0048)	0.2279 *** (0.0186)
N	270	270	270
AR(1)	0.0001	-	-
AR(2)	0.0760	-	-
Sargan	0.8665	-	-
Hansen J	0.3490	-	-
R^2^	-	0.6959	0.5809
F-statistics	-	744.09	4.37

Note: *, **, *** are expressed at 10%, 5%, and 1% levels respectively, the parentheses are t-statistic test values.

**Table 6 ijerph-19-15930-t006:** Regression analysis results under the instrumental variable method.

Variable	GMM	OLS	FE
*RPL*	0.2543 **	0.2591 ***	0.3039 **
	(6.2446)	(6.4548)	(2.9642)
*CM*	0.2034 *	0.1623 ***	1.1155 **
	(0.7354)	(0.7153)	(0.7076)
*SP*	0.0733 **	0.0629 *	0.0712 ***
	(2.2132)	(1.8538)	(1.6364)
*DRCE*	0.1841 ***	0.2345	0.0190 ***
	(1.2246)	(1.5242)	(0.1032)
*DT*	0.2645 ***	0.2467 **	0.0187
	(2.8469)	(2.8527)	(0.1667)
*PS*	0.1837 ***	0.1627 **	0.1143 **
	(3.2434)	(2.3504)	(1.9568)
CONS	0.792 ***	0.0218 **	0.0218 **
	(3.5634)	(1.9437)	(1.9437)
Observations	270	270	270
R-squared	0.6348	0.6255	0.7108
Wald-F stat	2674	2334	1192
*p*-value	0.3257	0.8112	0.3136

Note: *, **, *** are expressed at 10%, 5%, and 1% levels respectively, the parentheses are t-statistic test values.

**Table 7 ijerph-19-15930-t007:** Robustness test.

Variable	(1)	(3)	(4)	(5)	(6)
	The Decline in Total Carbon Emissions	Tax Relief	Government Investment	One-Stage Lag	Two-Period Lag
*TIAR*	0.1021 **	0.1034 **	0.0846 *	0.1235 ***	0.126 *
	(1.1254)	(2.2146)	(1.2146)	(1.7434)	(1.1421)
*CM*	0.1109 *	0.1141 **	0.0896*	0.1134*	0.5243**
	(2.8241)	(3.0525)	(1.4053)	(2.0291)	(4.1321)
*RI*	0.1139 *	0.0886 *	0.1427 **	0.0837 *	0.0796
	(2.1563)	(1.0653)	(2.1736)	(2.8791)	(1.2888)
*BTI*	0.0946 ***	0.1638 **	0.1684 *	0.0974 *	0.1018 **
	(1.1053)	(2.2864)	(1.1954)	(4.5286)	(1.3437)
*DT*	0.1036 **	0.1468 **	0.1238 *	0.1138 *	0.1542**
	(2.1433)	(2.6582)	(1.1143)	(1.5466)	(1.3587)
*PS*	0.1124 *	0.1327 **	0.1624 *	0.1854 *	0.1358 **
	(1.0465)	(2.9744)	(1.2154)	(1.5226)	(1.3742)
Control variables	Yes	Yes	Yes	Yes	Yes
Individual effect	Yes	Yes	Yes	Yes	Yes
Time effect	Yes	Yes	Yes	Yes	Yes
Observations	270	270	270	270	270
R^2^	0.259	0.486	0.434	0.247	0.225

Note: *, **, *** are expressed at 10%, 5%, and 1% levels respectively, the parentheses are t-statistic test values.

**Table 8 ijerph-19-15930-t008:** Regression Analysis on Influencing Factors of Low Carbon Development of Electric Power Industry in Subregion.

Variable	Eastern Region	Central Region	Western Region
*TIAR*	0.1526 *** (0.0097)	0.1683 *** (0.0025)	0.1697 *** (0.0087)
*CM*	0.1683 *** (0.0013)	0.1571 *** (0.0094)	0.1526 ** (0.0217)
*RI*	0.1648 ** (0.0410)	0.1932 ** (0.0464)	0.1752 ** (0.0378)
*BTI*	0.1196 ** (0.0373)	0.1505 * (0.0748)	0.1263 ** (0.0416)
*DT*	0.2783 ** (0.0309)	0.1684 ** (0.0412)	0.1997 ** (0.0385)
*PS*	0.1974 ** (0.0342)	0.1683 ** (0.0218)	0.1775 ** (0.0196)
*CONS*	0.1627 ** (0.0321)	0.15748 ** (0.0127)	0.1697 * (0.0841)
N	110	80	110
AR(1)	0.0002	0.0001	0.0000
AR(2)	0.0526	0.0612	0.0538
Sargan	0.8274	0.8873	0.8426
Hansen J	0.3257	0.3522	0.3327

Note: *, **, *** are expressed at 10%, 5%, and 1% levels respectively, the parentheses are t-statistic test values.

## Data Availability

China Comprehensive Energy Balance Sheet: Terminal Consumption. Available online: https://www.ceicdata.com/zh-hans/china/energy-balance-sheet/cn-energy-final-consumption (accessed on 27 November 2022). China Energy Statistics Yearbook 2012-2020. (In Chinese). Available online: http://www.stats.gov.cn/tjsj/tjcbw/202103/t20210329_1815748.html (accessed on 27 November 2022).

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
