# Peer review of "Influencing Factors and Realization Path of Power Decarbonization—Based on Panel Data Analysis of 30 Provinces in China from 2011 to 2019"

_ijerph, 2022, doi:10.3390/ijerph192315930_

Round 1

Reviewer 1 Report

I have read the article with interest. The Authors are investigating a significant problem. They used the Analytic Network Process (ANP) method to calculate the index weight and measure the decarbonization level of the power industry in 30 provinces in China from 2011 to 2019. As mentioned in the text, 2011-2019 was the critical period of the low-carbon transformation of the power industry, reflecting the deepening influence of market mechanisms. Therefore, the Authors studied the low-carbon process of the power industry from 2011 to 2019 to further explore the realized path. However, I am proposing a few amendments, which, in my opinion, will make the text clearer.

The authors should clearly state the paper's aim and main contributions in the abstract and introduction. We can read about it in lines 74-86 (and probably also in 204-208 and 575-576), but this information should be placed in the introduction before the description of the paper structure (lines 67-74). 

An important article's weakness is the lack of a discussion in which the Authors would compare their results to those from other publications. The second weakness is the lack of publications from recent years, especially in the literature review. The third weakness is technical formatting, for example, the description of formula 3., which could be a smaller font in the legend below it, or unreadable cells in table 4.

In general, the manuscript is clear, relevant for the field and presented in a well-structured manner. The cited references are relevant, but I recommend increasing referred papers published within the last five years, especially in the literature section in lines 87-202. The article does not include self-citations. 

In my opinion, the manuscript is scientifically sound and the experimental design (figure 1.) is appropriate to construct the scenerio of path to realize the low-carbon development of the power industry. The manuscript’s results appear reproducible based on the details given in the methods section. Nevertheless, we cannot interview the experts again. I have a technical recommendation to describe equations 1. and 2., for example, in the legend below them. 

The figures and tables schemes are appropriate. They properly show the results and data, and they are easy to interpret and understand. In my opinion, the results are interpreted appropriately and consistently throughout the manuscript. But I strongly recommend combining (if possible) figures 3-7, for example into two separate figures. 

The conclusions should be more precise and consistent with the evidence and arguments presented in the 4. Results, and 5. Discussion sections. 

Reviewer 2 Report

Dear authors,

The study refers to an interesting topic as ombined with the characteristics and development trends of the power industry, the five dimensions of "Power generation decarbonization (SP)", "Energy utilization efficiency (EU)", "Supply chain decarbonization (SC)", and "Power grid decarbonization (PG)", and "the Trading system (TS)" are selected to construct an evaluation index system for the power decarbonization and identify the key factors. Despite the effort that has been put into the study and its potential to contribute to the research field, the review revealed several major and minor areas of improvement required to increase the quality of the study. Some suggestions are made to improve the current manuscript:

1. The content of the abstract needs to be polished. Please add some appropriate management implications to the abstract.

2. Originality: The objective and relevance of the work should be more precisely stated. This is particularly evident in the introduction and model analysis. Unfortunately, the introduction does not manage to show the research gap in a comprehensible way. Also in the model analysis the results are written down without really questioning them critically. Further analyses are necessary to better prepare and discuss the results and thus generate a clear added value. This is a crucial point and requires further improvement.

3. Relationship to Literature: A major criticism of literature analyses, which unfortunately also applies here, is its rather low relevance and value contribution to research and practice. The same and limited literature is continuously cited. Therefore, further analyses, discussion points or future fields of research have to be added to increase the originality. 

4. Methodology: This research is used to the Analytic Network Process (ANP) method calculate the index weight and measure the decarbonization level of the power industry in 30 provinces in China from 2011 to 2019. It is difficult to understand what academic contributions are made to the study. What is the model formulation specificity and excellence used? 

5. Results: It is understood that the model of the analysis has been done properly. The conclusion section is too brief. An in-depth discussion should be given to support the purpose of the research.

6. Implications for research, practice and/or society: This study has some limitation, In this paper, the research scenario is defined as the construction of a new type of power system in China with the goal of low-carbon development. In the process of constructing the index system, some indicators are not selected. The contributions of this research are not obvious so author/s could improve implications clearly. That is, theoretical and practical implications based on the current research are weak in the conclusion section.

For all these reasons, I strongly suggest to major revise of this paper.

Round 2

Reviewer 1 Report

I recommend to publish the paper. Well done. 

I suggest only to further increase the readability of the graphical abstract.

Author Response

Thanks so much! We have gained a lot in the last round of amendments. and we have further modified the graphic abstract.

Reviewer 2 Report

Comments to the Author

Thank you for your revision. In general, this paper uses the Analytic Network Process (ANP) Method and Super Decision software to evaluate the effect of decarbonization in power industry. The paper identifies clearly between any implications for research, practice and/or society. It bridges the gap between theory and practice. In addition, these implications are consistent with the findings and conclusions of the paper. Limitations are properly drawn. To conclude, this paper contains new information adequate to justify publication.

I really enjoyed reading this well-written manuscript and I would recommend for publication consideration.

Author Response

Thank you very much! We have gained a lot in the last round of amendments, and we have further modified the graphic abstracte.